# Physiological stress in response to multitasking and work interruptions: Study protocol

Linda Becker[1]*, Helena C. Kaltenegger[2], Dennis Nowak[2], Matthias Weigl[2,3], Nicolas Rohleder[1]

1 Department of Psychology, Chair of Health Psychology, Friedrich-Alexander University Erlangen-Nürnberg, Erlangen, Germany, 2 Institute and Clinic for Occupational, Social and Environmental Medicine, LMU University Hospital Munich, Munich, Germany, 3 Institute for Patient Safety, University Hospital, Bonn, Germany

* linda.becker@fau.de

**Funding:** This study is part of the research project "Identifikation biomedizinischer und gesundheitlicher Wirkweisen von positiven und negativen Auswirkungen von digitalem Stress und dessen Bewältigung" [Identification of biomedical

## Abstract

### Background

The biopsychological response patterns to digital stress have been sparsely investigated so far. Important potential stressors in modern working environments due to increased digitalization are multitasking and work interruptions. In this study protocol, we present a protocol for a laboratory experiment, in which we will investigate the biopsychological stress response patterns to multitasking and work interruptions.

### Methods

In total, $N$ = 192 healthy, adult participants will be assigned to six experimental conditions in a randomized order (one single-task, three dual-task (two in parallel and one as interruption), one multitasking, and one passive control condition). Salivary alpha-amylase as well as heart rate as markers for Sympathetic Nervous System Activity, heart rate variability as measure for Parasympathetic Nervous System (PNS) activity, and cortisol as measure for activity of the hypothalamic-pituitary adrenal (HPA) axis will be assessed at six time points throughout the experimental session. Furthermore, inflammatory markers (i.e., IL-6, C-reactive protein (CRP), and secretory immunoglobulin-A) will be assessed before and after the task as well as 24 hours after it (IL-6 and CRP only). Main outcomes will be the time course of these physiological stress markers. Reactivity of these measures will be compared between the experimental conditions (dual-tasking, work interruptions, and multitasking) with the control conditions (single-tasking and passive control).

### Discussion

With this study protocol, we present a comprehensive experiment, which will enable an extensive investigation of physiological stress-responses to multitasking and work interruptions. Our planned study will contribute to a better understanding of physiological response patterns to modern (digital) stressors. Potential risks and limitations are discussed. The

and health effects of positive and negative effects of digital stress and coping with it] which is part of the Bavarian Research Association on Healthy Use of Digital Technologies and Media (ForDigitHealth), funded by the Bavarian Ministry of Science and Arts. Linda Becker has been partly funded by the Emerging Talents Initiative of the Friedrich-Alexander University Erlangen-Nürnberg. Matthias Weigl and Dennis Nowak have been partly funded by the Munich Centre for Health Sciences (MC-Health). We acknowledge support by Deutsche Forschungsgemeinschaft and Friedrich-Alexander-Universität Erlangen-Nürnberg (FAU) within the funding program Open Access Publishing. The funders had and will not have a role in study design, data collection and analysis, decision to publish, or preparation of the manuscript.

**Competing interests:** The authors have declared that no competing interests exist.

findings will have important implications, especially in the context of digital health in modern working and living environments.

# Introduction

## Rationale

Stress is one important factor influencing human health [e.g., 1,2]. Stress is part of everyday private and working life, is experienced by almost everyone, and is increasingly having an impact on health and life expectancy [3,4]. In modern, technology-driven working and living environments, new potential stressors related to digitalization (i.e., digital stressors) are more and more present. In the following, we will–similar to the concepts of techno strain and tech-nostress [5,6]–refer to stress that is related to the usage of digital technology and media as digital stress. Important potential digital stressors are multitasking and work interruptions [e.g., due to flooding text messages or emails; 7–11]. Both can be perceived as stressful and overwhelming [12–14] and cannot be avoided in many situations. However, this does not necessarily indicate that the feeling of being stressed or overwhelmed when being faced with these demands is also associated with a physiological stress response [15,16]. Although the biopsychological effects of several acute [e.g., social evaluation; 17] and chronic psychosocial stressors [e.g., caregiving; 18,19] and psychological determinants of biological stress-response patterns in general are well understood, only few attempts have been made so far to use this knowledge to understand the effects of stressors such as multitasking and work interruptions on biological stress system-activity. The aim of our planned study is to close this research gap.

The interpretation of a situation as threatening activates stress centers in the brain, which use stress systems to prepare the entire organism for dealing with the situation [20]. The Sympathetic Nervous System (SNS) activates systems throughout the body through noradrenergic innervation, which leads to the release of epinephrine and norepinephrine from the adrenal medulla and, among other things, results in an increase in heart rate and blood pressure [20,21]. The up-regulation of the SNS is accompanied by a down-regulation of the Parasympathetic Nervous System [PNS; 1,21]. The slower response of the hypothalamic-pituitary adrenal (HPA) axis modulates the effects of the SNS and PNS by releasing the stress hormone cortisol from the adrenal cortex [21,22]. With a further delay [e.g., about 1.5–2 hours for interleukin-6 (IL-6); 23,24], complex effects of the immune system are activated with up-regulation of some components (most importantly inflammatory pathways) and down-regulation of others [most importantly cellular immunity; 2,25]. Temporarily, all these physiological stress responses are adaptive. Potentially harmful consequences arise when stress becomes chronic, i.e., when long-term stress exposure occurs [e.g., 19,26–28], or when so-called maladaptive stress-response patterns are used [e.g., 29–31]. We refer to stress-responses patterns as being maladaptive–in contrast to adaptive [e.g., 32,33]–when they do not allow the organism to efficiently cope with or to adjust the individual's physiological responses or behavior to the situation.

With regard to stress effects on health, SNS, PNS, and HPA axis interact with patho-physiologically relevant systems, of which the inflammatory system is seen–beside e.g., glucocorticoids [34]–as one key factor [35]. Inflammatory processes are one of the central mechanisms in mediating the negative effects of stress on health [35]. Ultimately, acute stress exposure leads to systemic low-grade inflammation, which is–in the long-term–a key factor for the development of the most important diseases in industrialized nations such as cardiovascular

diseases, type-2 diabetes, and cancer [35,36]. Moreover, these patho-physiological stress-related processes are associated with a large number of other diseases such as chronic dermatological conditions (e.g., skin aging [37], urticaria [38,39], or skin tumors; [40]), asthma bronchiale [41,42], or obesity [43,44], and many more.

Although physiological responses can be triggered by stressors in principle, the actual stress response is associated with the nature of the stressor [so-called specificity hypothesis; 45,46]. According to this hypothesis, specifically situations that are perceived as threatening in contrast to challenging trigger HPA axis responses. Moreover, situations which are shameful or in which the social self is devaluated are associated with strong HPA responses [so-called social self-preservation theory; 47–49]. For cognitive stressors, both SNS and HPA axis responses have been reported [50,51], depending on task difficulty and on the presence of further stressors [52]. In a recent systematic review and meta-analysis from our group, which is currently under review [53], we found that SNS activity is significantly higher and PNS activity is significantly lower during dual- or multitasking than during single tasking. We identified no associations with HPA axis activity. However, the number of studies in which HPA axis reactivity to dual- or multitasking was investigated was small. We found no eligible studies in which immune system (re-) activity was investigated.

To summarize, so far, laboratory experiments in which the potential stressors multitasking and work interruptions are systematically induced are rare. Moreover, they vary in their potential how the biological stress systems are modulated and investigated. Multitasking and work interruptions differ from commonly investigated stressors in their nature as they primarily include a cognitive component in contrast to a psychosocial one, especially when induced digitally (i.e., without the presence of further persons). Therefore, with regard to the specificity hypothesis, it remains an open question whether physiological stress responses to multitasking and work interruptions differ between digital and non-digital stressors. In our planned study, we will therefore, differentiate between pure digital multitasking as well as work interruptions and comparable tasks in which another person is involved.

## Objectives

While the biopsychological effects of acute [e.g., social evaluation; 54] and chronic [e.g., caregiving; 18,19] psychosocial stressors and psychological determinants of biological stress-response patterns in general are well understood [e.g., 55,56], only few attempts have been made so far to use this knowledge to understand the effects of two prominent stressors in modern, digitalized working environments (i.e., multitasking and work interruptions) on biological stress systems. With this protocol, we present a study plan for a comprehensive experiment, in which responses of the SNS, PNS, HPA axis, as well as of the immune system to multitasking and work interruptions will be investigated in a controlled laboratory setting.

Our primary research questions are:

1. Do dual- and multitasking conditions lead to physiological stress responses in comparison to a single-task control condition or a passive control condition?

2. Do work interruptions lead to physiological stress responses in comparison to a single-task control condition or a passive control condition?

3. Do the stress response patterns differ between digital and non-digital stressors?
   Additionally, the following secondary research question will be investigated exploratorily:

4. Do dual- and multitasking lead to perceived stress responses in comparison to a single-task control condition or a passive control condition?

5. Is the physiological stress response to dual- or multitasking, or work interruptions associated with person characteristics (e.g., age, sex, body-mass-index (BMI), education) and psychological variables (e.g., personality, coping style, self-efficacy, depression, anxiety, preference for multitasking [so-called polychronicity; 57]).

Because the associations with these factors will be investigated exploratorily (i.e., without specific hypotheses), only a subset, which is known to be highly relevant in stress research (i.e., age, sex, BMI), will be considered as covariates in the statistical analyses. The actual hypotheses are specified in the Materials and Methods section below.

## Materials and methods

### Study design

The study design is a cross-sectional laboratory experiment, in which participants will be randomly assigned to one of six conditions (four experimental conditions and two control conditions; Table 1; Fig 1).

### Participants

In total, $N$ = 192 (32 per condition) healthy, German-speaking adults between 18 and 40 years will be recruited. Exclusion criteria are diagnosed physical or psychological disorders (operationalized as diagnosed within the last 2 years), regular medication intake (exception oral contraceptives), pregnancy, a BMI $\geq$ 35 kg/m$^2$, regular smoking (more than 5 cigarettes per week), or being an employee of the Friedrich-Alexander University Erlangen-Nürnberg (FAU). The latter is a requirement from the workers' council of the FAU.

### Ethics approval

The study will be conducted according to the principles expressed in the Declaration of Helsinki and has been approved by the local ethics committee of the FAU (protocol number: 397_19 B).

### Power analysis

An a-priori power analysis has been conducted using G*Power (version 3.1.9.6). We expect to find small, but relevant effects and therefore used an effect size of $f$ = 0.14 for power calculation ($f^2$ = 0.02). We intend to achieve a power of 1 – β = 0.95 and used a Bonferroni-adjusted α-

**Table 1. Tasks that will be used in the six experimental conditions.**

| No. | Condition | Task 1 | Task 2 | Task 3 |
|---|---|---|---|---|
| 1 | Passive control (digital) | Watching a non-stressful video | / | / |
| 2 | Digital single-task (active control) | Continuous-performance task | / | / |
| 3 | Dual-task with digital interruptions | Continuous-performance task | Answering questions that are digitally presented as interruptions on the same monitor | / |
| 4 | Dual-task, parallel (digital) | Continuous-performance task | Answering questions that are digitally presented in parallel on a second monitor | / |
| 5 | Dual-task, parallel (non-digital) | Continuous-performance task | Verbal-fluency task (non-digital) | / |
| 6 | Multitask, parallel dual-task with additional non-digital parallel task | Continuous-performance task | Answering questions that are digitally presented in parallel on a second monitor | Verbal-fluency task (non-digital) |

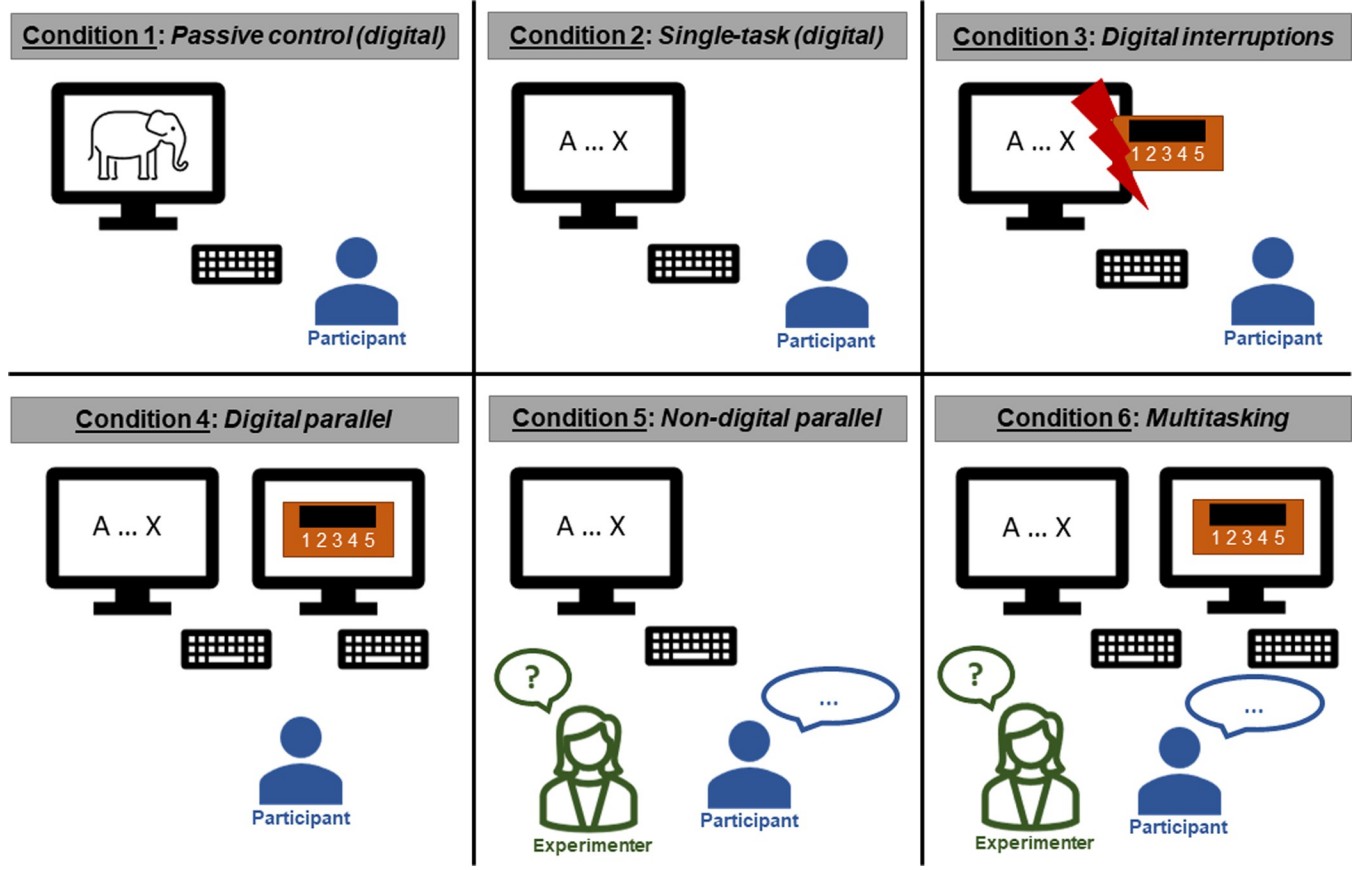

**Fig 1. Experimental conditions.** Each participant will be randomly assigned to one of the six experimental conditions. In condition 5 and 6, an experimenter will be present in the same room as the participant to introduce a non-digital, social-evaluative stress component.

level of $\alpha_{adjusted} = 0.05/3 = 0.017$ (see data preparation and statistical analysis). With these parameters a total sample size of $N = 174$ (29 per condition) is needed for an analysis of variance for repeated measurements (rmANOVA) with six measurement time points and six conditions. Based on our experience with similar studies, we expect a small drop-out rate of about 10% due to insufficient saliva or blood volumes and will, therefore, recruit additional 18 participants (3 per condition). This results in a total sample size of $N = 192$.

## Status and timeline of the study

Recruitment has started in August 2021 and data collection has started at the end of September 2021. The first version of this study protocol has been submitted before start of data collection. All recommendations from the reviewers' first reviews were considered. Data collection will last until the recruitment goal is fulfilled. The actual duration will depend on the development of the Covid-19 pandemic but is intended to last about one year. Data analysis will start after all data will have been collected. Preliminary analyses are not planned.

## Experimental conditions

An overview about all experimental conditions is provided in Table 1. Condition 1 will be a passive control condition, in which participants will watch a non-stressful video. Condition 2 will be an active control condition, in which participants will conduct a digital single-task,

which is the primary task in the other (dual- and multitasking as well as the interruption) experimental conditions. In conditions 3 to 5, participants will conduct the primary task either in combination with a parallel secondary task or will be interrupted by a secondary task. Participants who are assigned to condition 6 will be engaged in multitasking, which will include three tasks, the primary task as well as a digital and a non-digital parallel task. All conditions are visualized in Fig 1 and are summarized in Table 1.

## Tasks

**Digital primary task.** A computerized continuous-performance task [CPT; 58,59] will be the primary task for the single-task as well as the dual-task and multitasking conditions (conditions 2–6 in Fig 1). The CPT is a measure for sustained attention [60]. We will use an AX-CPT variant [e.g., 61], i.e. that the target is the letter 'X' occurring after an 'A'. In our version (Fig 2), the letters A, B, X, and Y will be presented, and a response button should be pressed after every second letter, i.e., after a pair of letters. The cue letter is 'A' and the target letter is 'X', i.e., that one response button should be pressed after each A-X pair. In all other cases (after A-Y, B-X, B-Y pairs), the other response button should be pressed (Fig 2). In our AX-CPT version, the letter presentation time is 2,000 ms and the inter-stimulus interval is 3,500 ms. The probability of an A-X pair is 0.50 and 0.17 for the other pairs.

**Digital secondary task.** As a digital secondary task, questions and five answer possibilities, of which one is correct, will be presented to the participants. The answer possibilities will be assigned to five answer buttons, and the instruction is that the correct answer should be chosen as fast as possible. As questions, items from the *Intelligenz-Struktur-Test 2000R* [IST-2000R; 62,63] will be used and will be presented in a randomized order. The following sub-tasks will be chosen: sentence addition (*Satzergänzung*), analogies (*Analogien*), and figure selection (*Figurenauswahl*). In condition 3 (digital interruptions), the primary task will be interrupted by questions from the IST 2000R, which will appear on the same monitor. The primary task will stop and will be overlayed by the secondary one during the presentation of the secondary task and will automatically continue after a response is given by the participant. In condition 4 (dual-task, parallel condition), participants will have to perform the IST 2000R on a separate computer screen parallel to the primary task. The participants will be instructed that both tasks are equally relevant, i.e., that none of the tasks should be prioritized, and that both tasks should be performed as accurately as possible.

**Non-digital secondary task.** As non-digital secondary task, participants will have to conduct a verbal-fluency task [VFT; e.g., 50,64]. The instruction, which will be verbally given by the experimenter, is to name as many words as possible that belong to a given category or which begin with a given letter. The time per category/letter will be 2 minutes, and time between two rounds will be 1 minute. Each participant will conduct six rounds with the same categories/letters in a randomized order. This secondary task will be conducted in parallel to the primary task (i.e., the AX-CPT; condition 5 in Fig 1). The experimenter will be in the same room as the participant to introduce a social-evaluative component. As for the digital parallel task, participants will be instructed that both tasks are equally relevant, i.e., that none of the tasks should be prioritized, and that both tasks should be performed as accurately as possible.

In the multitasking condition (condition 6 in Fig 1), all three tasks (AX-CPT, the digital parallel task, as well as the VFT) should be performed in parallel and, again, none of the tasks should be prioritized.

**Passive control condition.** The (digital) passive control condition will be to watch a non-stressful documentary video (condition 1 in Fig 1). The content of the video is landscapes and

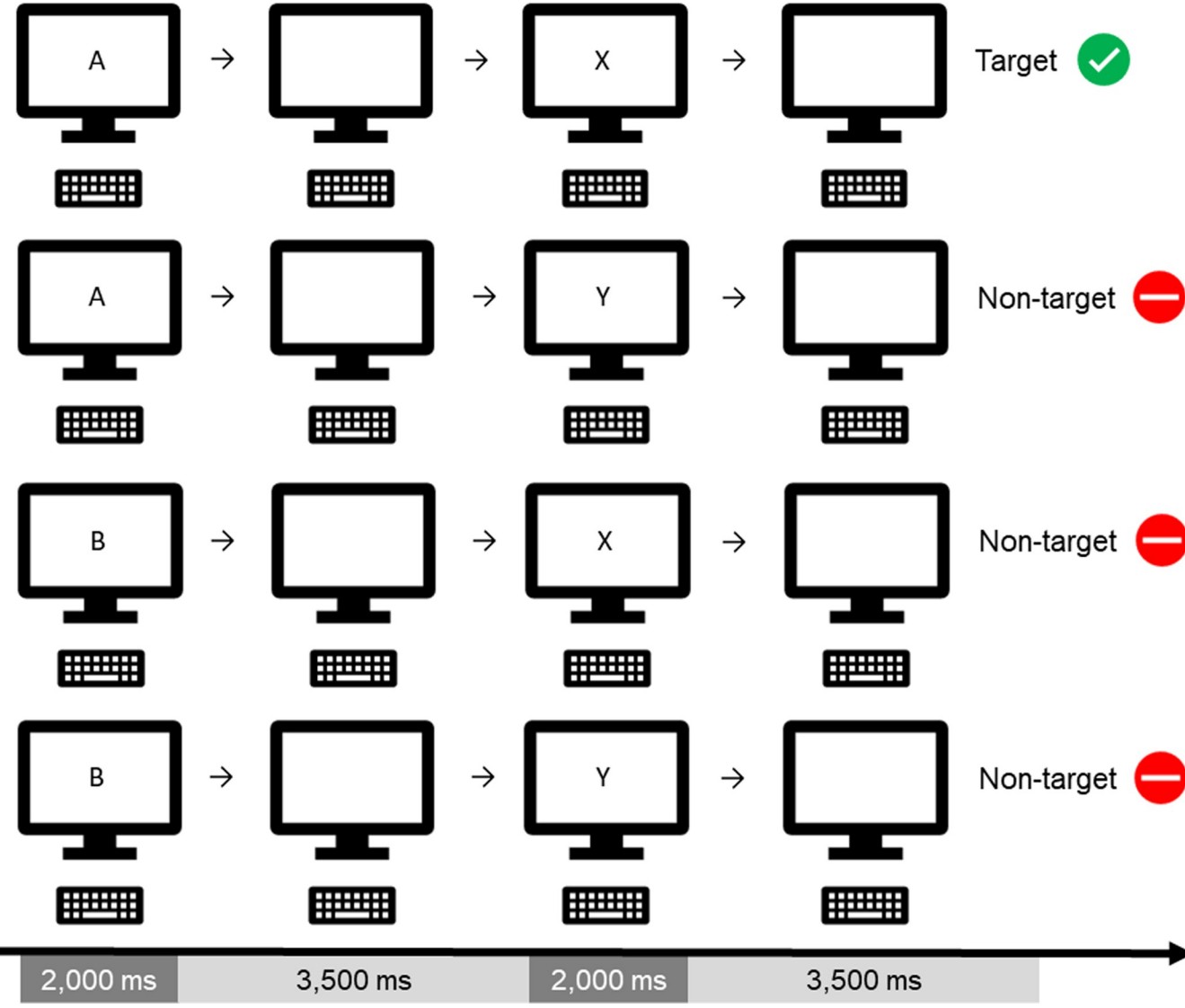

**Fig 2. AX-CPT.** In our version of the AX-CPT, the target pair will be A-X. A-Y, B-X, as well as B-Y pairs will be non-targets. This task will be used as the digital single-task and will be the primary task in the dual- and multitasking conditions.

animals. These videos have been successfully used in previous studies and evaluated as being non-arousing [65,66].

### Biological and physiological outcome measures

Main outcomes will be measures for physiological stress responses, which will be assessed via saliva and capillary blood samples as well as by means of electrocardiogram (ECG) recordings. An overview is provided in Table 2. As measures for SNS re-activity, salivary alpha-amylase [67] as well as the participants' heart rate will be used. PNS re-activity will be assessed from heart rate variability (HRV) measures [e.g., root mean square of successive differences of consecutive R-R intervals (RMSSD); 68]. Activity of the HPA axis will be assessed via salivary cortisol measurements, which will be collected by means of *Salivettes* (Sarstedt, Nümbrecht, Germany). IL-6 and CRP will be assessed from capillary blood samples, which will be

**Table 2. Biological and physiological outcome measures, number of samples, collection time point, and assessment procedure.**

| Stress system | Outcome measure | Number of time points | Time points | Procedure |
|---|---|---|---|---|
| Sympathetic Nervous System | Salivary-alpha amylase | 6 | Baseline, +0, +10, +20, +45, +90 minutes | Stimulated saliva samples |
| | Heart rate | NA | Continuously (baseline to + 90 minutes) | Electrocardiogram |
| Parasympathetic Nervous System | Heart rate variability (RMSSD) | NA | Continuously (baseline to + 90 minutes) | Electrocardiogram |
| HPA axis | Cortisol | 6 | Baseline, +0, +10, +20, +45, +90 minutes | Stimulated saliva samples |
| Immune system | CRP | 3 | Baseline, + 90 minutes, +24 hours | Dried Blood Spots |
| | IL-6 | 3 | Baseline, + 90 minutes, +24 hours | Dried Blood Spots |
| | s-IgA | 2 | Baseline, +90 minutes | Unstimulated saliva samples |

*Note*. CRP: C-reactive protein; HPA: hypothalamic-pituitary adrenal; IL-6: interleukin-6; s-IgA: secretory immunoglobulin-A; NA: not applicable; RMSSD: root mean square of successive differences.

measured in Dried Blood Spots [69,70], which is an established procedure in our group [e.g., 71,72]. s-IgA will be measured in unstimulated saliva samples, which will be collected by means of *Salicaps* (IBL international, Hamburg, Germany).

## Assessment of sample characteristics

Demographic, anthropometric, socio-economic, lifestyle, as well as health-related variables will collected via questionnaires (e.g., age, sex, ethnicity, education, occupation, mother tongue, smoking status, diseases, medication intake). Some of these characteristics (e.g., smoking status and mother tongue) will be only collected to double-check whether inclusion criteria are fulfilled. A full item list is provided as Supplementary Material in S1 File.

## Stress perception, affect and anxiety during the experiment

During the experimental session, stress perception, anxiety, and affect as measures for the perceived (non-physiological) stress response will be assessed via the following instruments:

**Affect.**  Positive and negative affect will be measured four times during the experiment by means of the Positive and Negative Affect Schedule [PANAS; 73,74]. Assessment time points will be before the task, immediately after it, 20 minutes after the task, as well as 90 minutes later.

**State anxiety.**  State anxiety and depression before and immediately after the task will be measured by using the state items from the State-Trait Anxiety-Depression Inventory [STADI-S; 75].

**Stress perception.**  Perceived stress will be assessed throughout the experiment at the time points of the saliva samples by means of 10-point Likert scales with the anchors "not stressed at all" and "extremely stressed". This scale has been successfully used in previous studies from our group [e.g., 76–79]. Additionally, participants are asked about their perceived exertion as well as their level of tiredness, again on 10-point Likert scales. All three scales are provided as Supplementary Material in S2 File.

## Psychological variables

Beside demographic, anthropometric, and health -related variables (see above), the acute stress response is related with a variety of psychological variables [e.g., 55]. Therefore, psychological variables as well as lifestyle factors, which might be related with the stress responses under study, will be assessed additionally. The following standardized questionnaires or in-house developed items will be used.

**Burnout.**    The Maslach Burnout Inventory [MBI; 80] will be used for assessment of burn-out symptoms.

**Coping.**    Coping will be assessed by means of the German 24-item version of the Coping Inventory for Stressful Situations [CISS; 81,82]. The scale assesses task-oriented, emotion-focused, as well as avoidance-oriented coping. The avoidance-oriented coping scale can be further divided into distraction coping and social-diversion coping subscales.

**Depression.**    Depression will be assessed by means of the German version of the long form of the depression scale from the Center for Epidemiological Studies [CES-D; 83,84].

**Emotion regulation.**    A German version of the Emotion Regulation Questionnaire [ERQ; 85,86] will be used to assess the emotion regulation dimensions reappraisal and suppression.

**Habitual multimedia consumption.**    For assessment of habitual television and internet usage, items from a questionnaire for the assessment of habitual media consumption by Koch [2010; 87] will be used. Each subscale consists of 8 items that should be answered on 7-point Likert-scales. A further scale, which assesses habitual mobile phone usage, has been developed analogously based on the items by Koch [2010, 87] for habitual television and internet usage.

**Multimedia usage.**    Self-developed items will be used to assess time of day at which media is used, separately for weekdays and weekends. A German version of the Media and Technology Usage and Attitudes Scale [MTUAS; 88] will be used to assess frequency of multimedia usage, social media activity, as well as attitudes towards media usage. The polychronicity (i.e., the preference for multitasking) items from the MTUAS will be left out. Polychronicity will be assessed by means of the Multitasking Preference Inventory [MPI; 57] instead.

**Multitasking.**    The amount and causes of multitasking behavior in everyday life will be assessed by means of 8 self-developed items, which will be answered on 5-point Likert Scales. The items are provided as Supplementary Material in S3 File.

**Perceived stress.**    Perceived stress during the last month will be assessed by means of a German translation of the 10-item version of the Perceived Stress Scale [PSS; 89,90].

**Personality.**    For personality assessment, the short version of the Big Five Inventory [BFI-K; 91] will be used. This questionnaire enables assessment of the big-five personality-dimensions extraversion, neuroticism, agreeableness, conscientiousness, and openness to experience by means of 21 items.

**Resilience.**    For assessment of resilience, the German 11-item resilience scale [RS-11; 92] will be used.

**Self-efficacy.**    Self-efficacy will be assessed by means of a German scale for general self-efficacy [SWE; 93]. The scale includes ten items, with which self-efficacy is rated on 4-point Likert scales.

**Social anxiety.**    Social anxiety will be assessed by means of the Social Interaction Anxiety Scale (SIAS) as well as the Social Phobia Scale [SPS; 94].

**Social support.**    The 14-item version of the *Fragebogen zur sozialen Unterstützung* [FSo-zU-K14; 95] will be used to assess social support.

**Trait anxiety.**    Trait anxiety and depression will assessed by means of the trait-items from the State-Trait Anxiety-Depression Inventory [STADI-T; 75]. Note that we will use the CES-D as main outcome measure for depression. However, we will leave in the depression items to not alter the psychometric properties of the STADI-T.

## Experimental setting and procedure

After providing informed and written consent, participants will be equipped with the heart rate monitors and will be familiarized with the saliva collection procedure via Salivettes (i.e., the collection of the stimulated saliva samples), and a practice saliva sample ($s_0$) will be

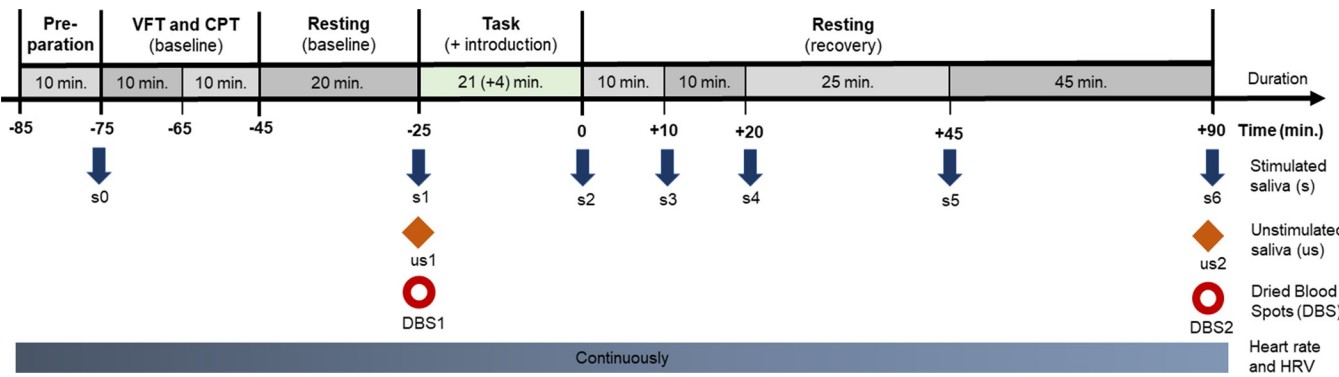

**Fig 3. Timeline of the experiment.** The entire session will last about 3 hours. A further Dried Blood Spot Sample (DBS$_3$) will be collected 24 hours after DBS$_1$ (not shown). *Note*. CPT: continuous performance task; DBS$_i$: Dried Blood Spot #$i$; HRV: heart rate variability; $s_i$: stimulated saliva sample #$i$, which will be collected by means of Salivettes; us$_i$: unstimulated saliva sample #$i$; VFT: verbal-fluency task.

collected, which will not be used for later analyses. After this, all participants–irrespective of the actual group assignment–will conduct practice trials of the VFT and the CPT and will be informed that they will possibly have to repeat these tasks throughout the session. This will be followed by a resting period of about 20 minutes. At the end of the resting period, the first blood spot sample (DBS$_1$) and the next stimulated saliva sample ($s_1$) as well as the first unstimulated saliva sample (us$_1$) will be collected. After this, the main task will be introduced to the participants and will then start immediately and will last about 21 minutes. Introduction of the task and the actual task will take a total of about 25 minutes. Immediately after the end of the main task, participants will provide the second saliva sample ($s_2$; i.e., end of the main task +0 minutes). The subsequent stimulated saliva samples will be collected at the following time points: +10 ($s_3$), +20 ($s_4$), +45 ($s_5$), and +90 ($s_6$) minutes after the end of the main task. The second Dried Blood Spot (DBS$_2$) as well as the second unstimulated saliva sample (us$_2$) will also be collected at + 90 minutes post task. During the time window between the task and the last saliva and blood sample, participants will rest and will fill out questionnaires until the end of the session and will be allowed to take as many breaks as they prefer. The goal of this period is not to stress the participants while conducting a non-stressful task (i.e., filling-out the questionnaires). The timeline of the whole experiment is shown in Fig 3. The entire session will last about 3 hours. Additionally, participants will provide another blood sample (DBS$_3$) 24 hours after the first one. This DBS collection will be conducted by the participants themselves at their homes and samples will be send back via mail.

## Outcome measures

**Primary outcomes.** Primary outcomes will be the physiological stress measures for SNS, PNS, HPA axis, and immune system reactivity (i.e., sAA, HR, RMSSD, cortisol, CRP, s-IgA, and IL-6).

**Secondary outcomes.** Secondary outcomes will be subjective stress perception, state anxiety, and affect as well as associations between the time course of the primary outcomes with person characteristics (e.g., age and sex) and psychological variables (e.g., personality, coping style, self-efficacy, depression, anxiety, polychronicity).

## Hypotheses

**Main hypotheses.** With regard to the specificity hypothesis [45,46], we expect that pure digital stressors will trigger physiological responses of the SNS (i.e., an up-regulation), PNS

(i.e., a down-regulation), as well as an activation of the immune system, but not of the HPA axis (due to the absence of a social-evaluative component). An additional activation of the HPA axis (i.e., an increase of cortisol levels) is expected for the conditions 5 and 6 (non-digital parallel dual-tasking and multitasking), in which we included a social-evaluative component by the presence of an experimenter.

Our main hypotheses, that are associated with research questions 1, 2, and 3 are:

1. Conditions 3 and 4 (digital work interruptions and digital parallel dual-tasking) will trigger responses of the SNS, PNS, and the immune system that are stronger than in the passive control condition 1 and the single-task control condition 2. No HPA axis response are expected for conditions 3 and 4.

2. Conditions 5 and 6 (non-digital parallel dual-tasking and multitasking) will trigger responses of the SNS, PNS, HPA axis, and the immune system that are stronger than in the passive control condition 1 and the single-task control condition 2.

Regarding the time course of the physiological responses (if any will be found), we hypothesize fast responses of the SNS and PNS with a maximum immediately after the tasks. The HPA axis response is expected to be delayed with a maximum 20 minutes after the tasks. Slower responses are expected for the immune system with a maximum of 90 minutes after the task for IL-6 and s-IgA and 24 hours after the experimental session for CRP.

**Secondary hypotheses.**   We hypothesize that conditions 3, 4, 5, and 6 will induce the perception of being stressed and that perceived stress will be stronger in these conditions than in the passive control condition 1 and the single-task control condition 2 immediately after the stressor. Furthermore, we expect to find an increase in state anxiety and in negative affect as well as a decrease in positive affect immediately after the tasks, which will be stronger than in the passive control condition 1 and the single-task control condition 2. Moreover, we expect that the physiological stress response patterns will be associated with person characteristics [e.g., age and sex; e.g., 55] and psychological variables (e.g., personality, coping style, self-efficacy, depression, anxiety, and polychronicity).

## Sample handling and laboratory analysis of blood and saliva samples

The saliva and blood samples will be analyzed in our in-house laboratory (FAU, Chair of Health Psychology, Biopsychological Laboratory, Nürnberg, Germany) by trained staff using established procedures [e.g., 71]. After collection, Salivettes and Salicaps will be stored at -30˚C. On the analysis day, they will be thawed and centrifuged at 2,000 $g$ at 4˚C before further processing. The DBS samples will be dried for at least 8 hours at room temperature before they will be stored at -30˚C. The further handling during analysis will also be conducted according to established procedures [e.g., 69,71,96]. In short, a circle with a diameter of 3.5 mm will be punched out and will be eluted overnight in phosphate buffered saline which contains 0.1% Tween 20 solution. The next morning, samples will be shaken at 300 rpm for one hour before further processing.

Concentration of sAA will be measured with an enzyme kinetic assay, as described elsewhere [e.g., 97]. For salivary cortisol, CRP, as well as s-IgA measurement, high-sensitive Enzyme-linked Immunosorbant Assays [ELISA; e.g., 98,99] will be used. For IL-6 measurement, a ProQuantum-Immunoassay-Kit (ThermoFisher Scientific, USA) will be used. For determination of absolute CRP concentrations, linear regressions will be conducted which been validated in-house. sAA, cortisol, CRP, and s-IgA analyses will be conducted in duplicates and IL-6 analyses in triplicates. Analyses will be repeated if inter- or intra- coefficients of variation will be greater than 10%.

## Heart-rate variability analysis

Several HRV parameters can be derived from the ECG signal. The most prominent one, that is calculated in the time domain of the signal, is the RMSSD which is usually used as a marker for PNS activity [68]. The RMSSD will be the main HRV measure for our analyses and will be used for main hypotheses testing. Yet, there are further HRV parameters that can be extracted from the HRV signal, e.g., in the very-low frequency (VLF), low frequency (LF), and high-frequency (HF) range, which can be derived from the power spectrum of the HRV signal [68,77]. It has been suggested that VLF power is associated with SNS activity, LF power with both SNS and PNS activity, and HF power with PNS activity [68]. However, there remains a debate whether all these components in the frequency domain reflect parts of both, SNS and PNS activity [100]. Therefore, these components will not be included in our main analyses, but additional analyses will be conducted. Another frequently used HRV parameter as a measure for sympatho-vagal balance is the ratio LF/HF [101], which will also be used for additional analyses in our study. For all these additional HRV analyses (besides the RMSSD) an adjusted alpha-level of $\alpha_{adjusted\,=}$ .001 will be applied.

## Randomization

For randomization, computer-generated randomization lists will be used, which will be stored in closed envelopes and will be handed out to the experimenters immediately prior to the experimental sessions. A team member who is not involved in data collection is responsible for randomization. Subjects are blinded and are informed that the intention of the study is the assessment of physiological responses to interaction with digital devices. Participants' sex will be considered in the randomization process to ensure an equal sex distribution.

## Data preparation and statistical analysis

For statistical analysis, IBM SPSS Statistics (version 26 or higher for Windows) will be used. Data will be screened for outliers, and outliers which differ more than 3 standard deviations from the participants' mean will be excluded from further analysis. Test of normality will be conducted by means of the Kolmogorov-Smirnov test. If necessary, data will be transformed (e.g., by means of the natural logarithm or square root transformation) to achieve a normality distribution.

For main hypotheses testing (research questions 1, 2, and 3), rmANOVAs will be conducted. In the analyses, the between-subjects factor 'Condition' (with six levels, see above) as well as the within-subjects factor 'Time' (with either six levels (for sAA, cortisol, and ECG measures), two for CRP and IL-6, and two for s-IgA) will be included. Three separate analyses will be conducted, one for SNS and PNS markers (i.e., sAA, HR, and RMSSD) and six measurement time points ($s_1$ to $s_6$), one for cortisol as the only HPA-axis marker with the same six time points, and one for the immune parameters (i.e., IL-6, CRP, and s-IgA) and two measurement time points. An additional factor 'Measure' (e.g., with the levels IL-6, CRP, and s-IgA for immune parameters) will be included if necessary. A Bonferroni-adjusted $\alpha$-level of $\alpha_{adjusted} = 0.05/3 = 0.017$ will be used for the main analyses, because three separate rmANOVAs will be conducted. The potential confounders age, sex, BMI, use of oral contraceptives and menstrual cycle phase for female participants, as well as time of day will be included in all statistical analyses as covariates.

Research question 4 (i.e., perceived stress, state anxiety, and affect) will be investigated analogously to research questions 1 to 3 by means of a further rmANOVA. The same $\alpha_{adjusted} = 0.017$ as for main hypothesis testing will be used. For analysis of research question 5 (i.e., associations with person characteristics and psychological variables), nominal scaled variables

(e.g., sex and education) will be included as additional factors in further rmANOVAs. For metric variables, multivariate regression analyses will be conducted in which the potential moderator variables will be included as moderator variables. For these analyses, the SPSS macro PROCESS [102] will be used. For these analyses which refer to research question 5, an adjusted $\alpha$-level of $\alpha_{adjusted} = 0.001$ will be used.

### Data management plan

To protect the participants' privacy and to maintain confidentiality, all personal data is stored in password-protected files and secured against unauthorized access by third parties. The raw data and materials are only accessible to project-team members. Each participant is assigned a randomly generated code that does not allow any conclusions to be drawn about the person. Only this code is used for naming files and samples. Only completely anonymized data will be made available to other researchers after completion of the study or in data repositories. Only mean values and group statistics will be reported in publications.

### Dissemination and analysis plans

Data analyses will start after data collection is completed. Interim analyses are not planned. At least one paper will be submitted to a leading journal in the field. The pre-processed and anonymized data will be made publicly available at the Open Science Framework (OSF) after an Embargo period of about 5 years.

## Discussion

### Summary

Psychosocial stress and its immediate and long-term biological effects have been studied relatively well, but a link between psychobiological-oriented stress research and the effects of digital stress has largely been lacking. The aim of our study is to apply the methods of psychobiological stress research in the context of digital stress (i.e., the widespread phenomena multitasking and work interruptions), with the overarching goal to provide the foundations for a better understanding of the health effects of digital stress. We describe a study protocol of a comprehensive experiment that investigates effects of multitasking and work interruptions on physiological stress response patterns including SNS, PNS, HPA axis, and the immune system.

### Strengths of the planned study

So far, acute psychobiological stress response patterns have been well researched in the context of "classic" stress research for non-digital stressors. In contrast, psychobiological stress reaction patterns in the context of digital stress have hardly been studied so far [103]. The aim of the presented study is to bring digital stress from every day and working life into the laboratory context, and thus to make it experimentally investigable. Our experimental approach is the key strength of our study. It enables us to systematically induce multitasking demands or work interruptions and compare them with single-task and passive control conditions.

Moreover, our study will enable to differentiate between stress responses to pure digital stressors and stressors that also include a non-digital component (operationalized by the presence of an experimenter). Nevertheless, all active tasks–the digital and the non-digital ones–include a cognitive component and therefore our study design does not allow to differentiate between digital and cognitive stressors. However, this is not a restriction of our study with

respect to our definition of digital stress as being related to the usage of digital technology and media, which is independent of the (e.g., mental) processes being involved.

## Potential risks

From a practical point of view, there is a risk that the blood and saliva volumes will be too low for analysis. However, this has been included in the sample size-calculation and will be minimized by training the experimenters by experienced researchers. A further potential risk is that the recruitment goal might not be fulfilled, e.g., due to the Covid-19 pandemic. In the case that another lockdown will be imposed, recruitment will be paused.

## Limitations

In our study, we will focus on multitasking and work interruptions as highly relevant modern stressors. Nevertheless, a variety of further digital stressors is conceivable [e.g., techno insecurity, techno overload, or techno invasion; 104]. Moreover, our operationalization of multitasking and work interruptions is just one of many possibilities and many more are possible.

A further limitation is that we cannot assess certain factors which might be related with the physiological stress response to multitasking and work interruptions. One of these are primary and secondary appraisal [105], which are known to be associated with acute stress responses in general. However, assessing them between the introduction of the task and the beginning of the task [e.g., by means of the Primary and Secondary Appraisal Scale; 106] would disrupt the procedure too much. A further and related factor which we cannot assess is executive functioning, of which especially attentional control has been shown to be related with HPA axis responses after acute stressors [107]. Furthermore, although we assess intelligence during some of the sub tasks, a comprehensive assessment [including emotional intelligence; 108] would be even more meaningful. A further potential limitation is the chosen passive control condition as we cannot rule out that watching the videos unintendedly leads to either stress induction or relaxation. However, the videos' contents have been rated as being low arousing in previous studies [65,66]. Moreover, this task is better suited than other potential control tasks in which participants are instructed to do nothing at all, which are–in our opinion–more likely to induce relaxation and which are also much more difficult to control.

We will use a pragmatic age restriction of 40 years, because this has been shown to be suitable in previous studies from our group for recruiting healthy participants who do meet all inclusion criteria. Other age groups (older than 40 years as well as children and adolescents) would be interesting target groups for future follow-up studies. Besides, non-healthy participants with diseases which are known to be associated with stress reactivity or being associated with chronic stress development [e.g., depression or post-traumatic stress disorder, 109] should be subject of further investigations.

## Implications

Nevertheless–despite these limitations–, we deem that our study contributes to a deeper understanding on influences of stressors associated with the use of digital technology on humans' health-outcomes. Specifically, our results are expected to expand our current knowledge base on the impact of multitasking and information load on humans' psychobiological stress responses with particular focus on physiological response patterns. Stress exerts a strong influence on health via well-described processes [e.g., conceptualized in the Allostatic Load Model; 3,32,110]. In the long-term, stress is negatively associated with quality of life, health, and longevity of individuals and, thus, productivity of society [111,112]. Therefore, our study is of high relevancy.

Given the ubiquitous applications of digital technologies in modern workplaces and living environments, our findings will help to further understand the mechanisms between digital stressors in various occupational settings and adverse health outcomes [113]. Eventually, examination of job-related risks will inform policy and practice interventions in occupational health. Overall, the findings from our study will have important implications for better understanding the long-term health effects of the potential stressors multitasking and work interruptions in several settings.

## Conclusions

Our planned study will expand our understanding of the physiological response patterns due to the modern (digital) stressors multitasking and work interruptions. By quantifying objective parameters of biological stress responses and, thus, patho-physiologically relevant markers of digital stress, the health effects of digital stress will be made assessable. The findings will have important implications, especially in the context of health in digitalized working environments.

## Supporting information

**S1 File. Questionnaire sample characteristics.** Questionnaire that will be used for the assessment of sample characteristics. In the actual study, a German version will be used.
(PDF)

**S2 File. Visual-analogous scales.** Visual-analogous scales with which perceived stress, tiredness, and exertion will be assessed. In the actual study, a German version will be used.
(PDF)

**S3 File. Multitasking questionnaire.** Questionnaire that will be used for the assessment of multitasking. In the actual study, a German version will be used.
(PDF)

## Acknowledgments

We thank Laura Carolina Manns for assisting the development of the multitasking items and Janek Ruß for supporting translation of the MTUAS. Furthermore, we thank Swathi Hassan Gangaraju for programming the tasks and Katharina Hahn for supporting setting-up the experiment.

## Author Contributions

**Conceptualization:** Linda Becker, Helena C. Kaltenegger, Dennis Nowak, Matthias Weigl, Nicolas Rohleder.

**Funding acquisition:** Dennis Nowak, Matthias Weigl, Nicolas Rohleder.

**Methodology:** Linda Becker, Helena C. Kaltenegger, Dennis Nowak, Matthias Weigl, Nicolas Rohleder.

**Project administration:** Linda Becker.

**Resources:** Nicolas Rohleder.

**Software:** Linda Becker.

**Supervision:** Linda Becker, Dennis Nowak, Matthias Weigl, Nicolas Rohleder.

**Visualization:** Linda Becker.

**Writing – original draft:** Linda Becker.

**Writing – review & editing:** Linda Becker, Helena C. Kaltenegger, Dennis Nowak, Matthias Weigl, Nicolas Rohleder.

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
