## [Decision Letter · Decision Letter 0]

1 Nov 2021

PONE-D-21-28194Physiological stress in response to multitasking and work interruptions: Study protocolPLOS ONE

Dear Authors,

Thank you for submitting your manuscript to PLOS ONE. After careful consideration, we feel that it has merit but does not fully meet PLOS ONE’s publication criteria as it currently stands. Therefore, we invite you to submit a revised version of the manuscript that addresses the points raised during the review process.

We look forward to receiving your revised manuscript.

Kind regards,

Eva M J Peters, M.D.

Academic Editor

PLOS ONE

Journal Requirements:

“This study is part of the research project "Identifikation biomedizinischer und gesundheitlicher Wirkweisen von positiven und negativen Auswirkungen von digitalem Stress und dessen Bewältigung“ [Identification of biomedical and health effects of positive and negative effects of digital stress and coping with it] which is part of the Bavarian Research Association on Healthy Use of Digital Technologies and Media (ForDigitHealth), funded by the Bavarian Ministry of Science and Arts. Linda Becker has been partly funded by the Emerging Talents Initiative of the Friedrich-Alexander University Erlangen-Nürnberg. Matthias Weigl and Dennis Nowak have been partly funded by the Munich Centre for Health Sciences (MC-Health). We acknowledge support by Deutsche Forschungsgemeinschaft and Friedrich-Alexander-Universität Erlangen-Nürnberg (FAU) within the funding program Open Access Publishing. The funders had and will not have a role in study design, data collection and analysis, decision to publish, or preparation of the manuscript.”

We note that you have provided additional information within the Funding Section. Please note that funding information should not appear in other areas of your manuscript. We will only publish funding information present in the Funding Statement section of the online submission form.

“This study is part of the research project "Identifikation biomedizinischer und gesundheitlicher Wirkweisen von positiven und negativen Auswirkungen von digitalem Stress und dessen Bewältigung“ [Identification of biomedical and health effects of positive and negative effects of digital stress and coping with it] which is part of the Bavarian Research Association on Healthy Use of Digital Technologies and Media (ForDigitHealth), funded by the Bavarian Ministry of Science and Arts. Linda Becker has been partly funded by the Emerging Talents Initiative of the Friedrich-Alexander University Erlangen-Nürnberg. Matthias Weigl and Dennis Nowak have been partly funded by the Munich Centre for Health Sciences (MC-Health). We acknowledge support by Deutsche Forschungsgemeinschaft and Friedrich-Alexander-Universität Erlangen-Nürnberg (FAU) within the funding program Open Access Publishing. The funders had and will not have a role in study design, data collection and analysis, decision to publish, or preparation of the manuscript.”

Additional Editor Comments:

Dear Authors,

your manuscript has been reviewed favourably. Please answer all reviewers comments and resubmit.

On behalve of PLOS One, yours, Eva Peters

Reviewers' comments:

Reviewer's Responses to Questions

**Comments to the Author**

1. Does the manuscript provide a valid rationale for the proposed study, with clearly identified and justified research questions?

Reviewer #1: Yes

Reviewer #2: Yes

2. Is the protocol technically sound and planned in a manner that will lead to a meaningful outcome and allow testing the stated hypotheses?

Reviewer #1: Yes

Reviewer #2: Yes

3. Is the methodology feasible and described in sufficient detail to allow the work to be replicable?

Reviewer #1: Yes

Reviewer #2: Yes

4. Have the authors described where all data underlying the findings will be made available when the study is complete?

Reviewer #1: Yes

Reviewer #2: No

5. Is the manuscript presented in an intelligible fashion and written in standard English?

Reviewer #1: Yes

Reviewer #2: Yes

6. Review Comments to the Author

You may also provide optional suggestions and comments to authors that they might find helpful in planning their study.

Reviewer #1: In their manuscript, Becker et al. present a study protocol to investigate stress-reactivity inducing effects of dual- and multitasking as well as work interruptions by means of a standardized laboratory paradigm. The paradigm comprises 4 experimental conditions (digital interruptions, dual tasking with digital parallel task, dual tasking with non-digital task, and multi-tasking) and 2 control conditions (passive digital control and single digital task). The primary task is a computerized continuous-performance task; the digital secondary task comprises digitally presented items from an intelligence test with five answer possibilities; the non-digital secondary task consists of a verbal fluency task in the presence of a human experimenter; and the multitasking condition comprises all three tasks at the same time. Physiological reactivity testing includes assessment of SAM and HPA axis parameter reactivity before and up to 90 min after stress cessation, as well as immune measures, on a state-of-the-art level. The repeated measurement of relevant psychological state measures complements the physiological assessment and allows for psychological reactivity testing. Moreover, assessment of relevant trait measures allows to identify potential psychological correlates of physiological reactivity.

The presented paradigm aims at closing a gap by extending the range of standardized laboratory stress induction protocols in order to specifically investigate multitasking and work interruptions as supposed stress-inducing elements of a today´s digitalized working environment.

The presented paradigm and the proposed evaluation study are timely, highly innovative, important, and methodologically sound. I did enjoy reading this well-written manuscript.

I have some minor comments.

Introduction:

- I did miss a part where the authors explain the term digital stress. In line 55, the authors jump to multitasking and work interruptions as forms of stressors due to increased digitalization, but the term digital stress had not been introduced before. A prior definition/explanation/clarification of the term digital stress would be helpful for the reader.

- I would appreciate more careful wordings:

o Line 53: I totally agree that stress is an important factor influencing human health, but it is a bit much to state that it is one of the most important factors.

o “Most important” e.g. in lines 23, 53, 56 – maybe a more balanced wording?

o Line 83: I agree that inflammatory processes definitely play a role in mediating negative effects of stress on health, but data do not justify that inflammation is the one and only central mechanism for all existing negative effects of stress on health

- Line 90: only situations perceived as threatening? Or threatening and challenging as proposed by Lazarus in his transactional model of stress and shown in studies using the primary appraisal secondary appraisal (PASA) scale? Please also add references.

- Line 95: reference 36 has been submitted but not yet published – this should be pointed out more clearly

Objectives/Summary:

- Line 106 ff: In how far does “acute and chronic psychosocial stress” differ from “multitasking and work interruptions”? This is a bit confusing as there seems some overlap, especially since two of the six presented stress conditions comprise the experimenter as psychosocial stress element. What exactly is the difference - do you refer to non-digital acute stress tests comprising single tasks or consecutive single tasks as “acute and chronic psychosocial stress”? Please clarify and provide a clear conceptual differentiation between previous stress research and the present study.

- 445 ff: Similarly: in how far does “mental stress” differ from “digital stress”? It seems that mental stress includes digital stress as a sub form. Please clarify or correct.

- Line 115: dual and multitasking what? Conditions?

- Line 123: the physiological stress response to what?

Methods:

- There is some confusion with figure numbers; line 132 and 167 “Fig. 1” refers to Fig. 2; please check throughout the manuscript

- Line 191 ff:

o inconsistency – sometimes “condition”, sometimes “group” (l.199)

o There is some confusion between text and figure: “condition 2” (line 197) refers to conditions 3 in the figure; and “group 3” (line 199) refers to condition 4 in the figure. Line 325 “Fig. 3” refers to Fig 1. Please check and correct. AW: [ext] AG, aktueller Stand

- 215: please explicitly add (in brackets?) if this condition refers to condition 1 in the figure

- Please clarify: non-digital secondary task: how is the VFT instruction given - a written instruction? A verbal instruction provided by the experimenter?

- Please clarify: digital interruptions: does the primary task stop or does it continue (where – in the background?) when the interruption appears on the same screen?

- Line 276/277: what are “items from 64”? Please clarify.

- Why is depression assessed twice – CES-D and STADI-T?

- 353: typo: plural – triplicateS

- Randomization/group composition: The authors are experienced when it comes to effects of menstrual cycle phase and hormonal contraceptives on physiological stress reactivity. Please explain how you plan to rule out differential effects of female participants´ follicular and luteal phases of the menstrual cycle on (cortisol) stress reactivity and how to balance group compositions in terms of sex, use of hormonal contraceptives, and menstrual cycle phases.

Reviewer #2: Thank you for giving me the opportunity to review the study protocol regarding the study „Physiological stress in response to multitasking and work interruptions“. The study planned by Linda Becker and coworkers is of interest to the readership of PLOS in my opinion as it deals with a topic that affects many persons during digitalization, namely being confronted with many tasks at the computer at the same time. In the planned study the effects of single tasking, dual tasking and multiple tasking on physiological parameters will be assessed. I think the study proticol needs clarification in some points though. Here are my concerns:

1. Why do you assess IL-6 as this is not a typical stress marker and also you very clear state in the theoretical background that you expect stress effects on the immune systems with delay. I would suggest to measure IL-6, but rather not directly after the tasks.

2. Page 5, line 80: Please explain what maladaptibe stress response patterns are in your opinion.

3. Page 5, line 84: Please also give some examples on the relationship between stress and dermatological conditions as the relationship is well understood in this field.

4. Page 6, line 123: In case the physiological stress response ist associated with these person characteristics, they need to be considered as covariates! Please add!

5. Page 7, line 134: why do you use such an age restriction?

6. Page 7, line 135: How do you operationalize psychological disorders (ever during life or at the current moment?)

7. Page 7, line 136: Please consider to stratify groups for usage of oral contraceptives. I would prefer to do this instead of just statistically controlling for this factor.

8. Page 7, line 137: Why is being an employee of the University Erlangen-Nürnberg an exclusion criterion? I suggest to omit or explain further why this is necessary.

9. Has the study protocol been registered at DRKS? If not please do so.

10. Page 8, line 161: What is the content of the non-stressful video. How do you guarantee that the video does not induce relaxation? Would that be a problem? Please think about this and add to the discussion section.

11. Page 9, line 171: Please already explain at this point: Who will be involved in randomization? Are subjects blinded? If yes, what do they think the intention of the study is?

12. Page 9, line 178: please explain this abbreviation

13. Page 10, line 194: Is intelligence (IQ) used as covariate? Intelligent persons might not get frustrated so fast which can have an effect on the stress parameters. Please consider!

14. Page 11, line 225: IL-6: See above, as stated in the Theoretical background you do not really seem to expect short-term effects on IL-6 or do you? Please, be specific. Do you expect effects after 24 hours or also immediate effects?

15. Page 12, line 257: I suggest to come up with a more detailed figure showing the timeline and in which you illustrate when each variable will be assessed using different symbols for the different variables. Figure 1 is not quite understandable at one glance.

16. Page 13, line 264 and the following: what will you do with all these variables? Do you plan to use them as control variables? Are these dependent variables? If yes, which correction procedure (adjustment) will you use?

17. Page 17: What is your main heart parameter? It seems that you will exploratively look at everything regarding the heart rate variability. Please, explicitely state at this point whether you will correct for multiple testing or not.

18. It seems a bit odd to have a results section in study protocol, thus of a study which is still ongoing or has not even started yet. Please, omit this heading and include the necessary information of this section somewhere else (Methods Section or Theoretical background).

19. Has data collection started already?

20. Is it planned to conduct inbetween analyses?

21. Please add a data management plan. Where will the data be stored, who will have access, will the data be made available to other researchers or in data repositories? If not, please give an explanation.

22. What is the clinical relevance of the study? Please add to the discussion session.

7. PLOS authors have the option to publish the peer review history of their article (what does this mean?). If published, this will include your full peer review and any attached files.

Reviewer #1: No

Reviewer #2: No

---

## [Author Response · Author response to Decision Letter 0]

10 Nov 2021

Editor’s comments: 

Our response: We carefully checked the style requirements, made changes to the manuscript style, and renamed files whenever necessary. 

Our response: Data will be made available after an embargo period of about 5 years. Before the end of this period, it will be available upon request. Contact information has been included in the revised cover letter. 

Our response: Please see our response above. For the current manuscript (i.e., the study protocol), no data was recorded. Data of the planned study will be made available after an embargo period of about 5 years. Before the end of this period, it will be available upon request. 

“This study is part of the research project "Identifikation biomedizinischer und gesundheitlicher Wirkweisen von positiven und negativen Auswirkungen von digitalem Stress und dessen Bewältigung“ [Identification of biomedical and health effects of positive and negative effects of digital stress and coping with it] which is part of the Bavarian Research Association on Healthy Use of Digital Technologies and Media (ForDigitHealth), funded by the Bavarian Ministry of Science and Arts. Linda Becker has been partly funded by the Emerging Talents Initiative of the Friedrich-Alexander University Erlangen-Nürnberg. Matthias Weigl and Dennis Nowak have been partly funded by the Munich Centre for Health Sciences (MC-Health). We acknowledge support by Deutsche Forschungsgemeinschaft and Friedrich-Alexander-Universität Erlangen-Nürnberg (FAU) within the funding program Open Access Publishing. The funders had and will not have a role in study design, data collection and analysis, decision to publish, or preparation of the manuscript.”

We note that you have provided additional information within the Funding Section. Please note that funding information should not appear in other areas of your manuscript. We will only publish funding information present in the Funding Statement section of the online submission form.

“This study is part of the research project "Identifikation biomedizinischer und gesundheitlicher Wirkweisen von positiven und negativen Auswirkungen von digitalem Stress und dessen Bewältigung“ [Identification of biomedical and health effects of positive and negative effects of digital stress and coping with it] which is part of the Bavarian Research Association on Healthy Use of Digital Technologies and Media (ForDigitHealth), funded by the Bavarian Ministry of Science and Arts. Linda Becker has been partly funded by the Emerging Talents Initiative of the Friedrich-Alexander University Erlangen-Nürnberg. Matthias Weigl and Dennis Nowak have been partly funded by the Munich Centre for Health Sciences (MC-Health). We acknowledge support by Deutsche Forschungsgemeinschaft and Friedrich-Alexander-Universität Erlangen-Nürnberg (FAU) within the funding program Open Access Publishing. The funders had and will not have a role in study design, data collection and analysis, decision to publish, or preparation of the manuscript.”

Our response: We included the funding statement in the cover letter and removed it from the manuscript. 

Our response: Captions for the Supplementary Files have been included at the end of the revised manuscript.  

Reviewers' comments:

Reviewer #1: 

In their manuscript, Becker et al. present a study protocol to investigate stress-reactivity inducing effects of dual- and multitasking as well as work interruptions by means of a standardized laboratory paradigm. The paradigm comprises 4 experimental conditions (digital interruptions, dual tasking with digital parallel task, dual tasking with non-digital task, and multi-tasking) and 2 control conditions (passive digital control and single digital task). The primary task is a computerized continuous-performance task; the digital secondary task comprises digitally presented items from an intelligence test with five answer possibilities; the non-digital secondary task consists of a verbal fluency task in the presence of a human experimenter; and the multitasking condition comprises all three tasks at the same time. Physiological reactivity testing includes assessment of SAM and HPA axis parameter reactivity before and up to 90 min after stress cessation, as well as immune measures, on a state-of-the-art level. The repeated measurement of relevant psychological state measures complements the physiological assessment and allows for psychological reactivity testing. Moreover, assessment of relevant trait measures allows to identify potential psychological correlates of physiological reactivity.

The presented paradigm aims at closing a gap by extending the range of standardized laboratory stress induction protocols in order to specifically investigate multitasking and work interruptions as supposed stress-inducing elements of a today´s digitalized working environment.

The presented paradigm and the proposed evaluation study are timely, highly innovative, important, and methodologically sound. I did enjoy reading this well-written manuscript.

Our response: Thank you very much for the positive feedback and all your valuable suggestions, which helped us to further improve the manuscript. 

I have some minor comments.

Introduction:

- I did miss a part where the authors explain the term digital stress. In line 55, the authors jump to multitasking and work interruptions as forms of stressors due to increased digitalization, but the term digital stress had not been introduced before. A prior definition/explanation/clarification of the term digital stress would be helpful for the reader.

Our response: We included paragraph in the introduction in which we introduce the term digital stress: “In modern, technology-driven working and living environments, new potential stressors related to digitalization (i.e., digital stressors) are increasingly present. In the following, we will – similar to the concepts of techno strain and technostress (5,6) – refer to stress that is related to the usage of digital technology and media as digital stress.” (lines 52-56). 

- I would appreciate more careful wordings:

o Line 53: I totally agree that stress is an important factor influencing human health, but it is a bit much to state that it is one of the most important factors.

o “Most important” e.g. in lines 23, 53, 56 – maybe a more balanced wording?

Our response: The sentences have been revised to “Stress is one important factor influencing human health” (Abstract and line 50) and “Important potential digital stressors…” (line 56). 

o Line 83: I agree that inflammatory processes definitely play a role in mediating negative effects of stress on health, but data do not justify that inflammation is the one and only central mechanism for all existing negative effects of stress on health

Our response: We agree that inflammation is only one key factor and revised the paragraph accordingly: “…of which the inflammatory system is seen – beside e.g., glucocorticoids (34) – as one key factor (35). Inflammatory processes are one of the central mechanisms in mediating the negative effects of stress on health” (lines 86-88)

- Line 90: only situations perceived as threatening? Or threatening and challenging as proposed by Lazarus in his transactional model of stress and shown in studies using the primary appraisal secondary appraisal (PASA) scale? Please also add references.

Our response: In this paragraph, we refer to the social-self-preservation theory and the specificity hypothesis according to which HPA axis responses are especially elicited by threatening (rather than challenging) situations. The sentence has been revised to “According to this hypothesis, specifically situations that are perceived as threatening in contrast to challenging trigger HPA axis responses.” (lines 97-98). 

We carefully considered including the PASA in our study, but decided against it, because it would disrupt the procedure too much. Nevertheless, we now discuss this in the limitation section of our Discussion “A further limitation is that we cannot assess certain factors which might be related with the physiological stress response to multitasking and work interruptions. One of these are primary and secondary appraisal (104), which are known to be associated with acute stress responses in general. However, assessing them between the introduction of the task and the beginning of the task (e.g., by means of the Primary and Secondary Appraisal Scale; 105) would disrupt the procedure too much.” (lines 549-554). 

- Line 95: reference 36 has been submitted but not yet published – this should be pointed out more clearly

Our response: Unfortunately, the paper is still under review. We, therefore, point this out more clearly in the revised manuscript and added “…which is currently under review …” (lines 102-103). 

Objectives/Summary:

- Line 106 ff: In how far does “acute and chronic psychosocial stress” differ from “multitasking and work interruptions”? This is a bit confusing as there seems some overlap, especially since two of the six presented stress conditions comprise the experimenter as psychosocial stress element. What exactly is the difference - do you refer to non-digital acute stress tests comprising single tasks or consecutive single tasks as “acute and chronic psychosocial stress”? Please clarify and provide a clear conceptual differentiation between previous stress research and the present study.

Our response: We now provide examples for classic acute and chronic psychosocial stressors and discuss their differences and similarities with multitasking and work interruptions. The paragraph has been revised to “While the biopsychological effects of acute (e.g., social evaluation; 54) and chronic (e.g., caregiving; 18,19) psychosocial stressors and psychological determinants of biological stress-response patterns in general are well understood…” (lines 120-122). Furthermore, we added the following paragraph to the revised manuscript: “Multitasking and work interruptions differ from commonly investigated stressors in their nature as they primarily include a cognitive component in contrast to a psychosocial one, especially when induced digitally (i.e., without the presence of further persons). Therefore, with regard to the specificity hypothesis, it remains an open question whether physiological stress responses to multitasking and work interruptions differ between digital and non-digital stressors. In our planned study, we will therefore, differentiate between pure digital multitasking as well as work interruptions and comparable tasks in which another person is involved.” (lines 111-118). 

- 445 ff: Similarly: in how far does “mental stress” differ from “digital stress”? It seems that mental stress includes digital stress as a sub form. Please clarify or correct.

Our response: We revised the sentence to “Psychosocial stress and its immediate and long-term biological effects...” (line 512) and added the following paragraph to the Discussion section of the revised manuscript: “Moreover, our study will enable to differentiate between stress responses to pure digital stressors and stressors that also include a non-digital component (operationalized by the presence of an experimenter). Nevertheless, all active tasks – the digital and the non-digital ones – include a cognitive component and therefore our study design does not allow to differentiate between digital and cognitive stressors. However, this is not a restriction of our study with respect to our definition of digital stress as being related to the usage of digital technology and media, which is independent of the (e.g., mental) processes being involved.” (lines 530-536). 

- Line 115: dual and multitasking what? Conditions?

Our response: Research question 1 has been revised to: “Do dual- and multitasking conditions lead to physiological stress responses in comparison to a single-task control condition or a passive control condition” (lines 129-130). 

- Line 123: the physiological stress response to what?

Our response: Research question 5 has been revised to “Is the physiological stress response to dual- or multitasking, or work interruptions associated with person characteristics (e.g., age, sex, body-mass-index (BMI), education) and psychological variables (e.g., personality, coping style, self-efficacy, depression, anxiety, preference for multitasking (so-called polychronicity; 57).” (lines 137-140)

Methods:

- There is some confusion with figure numbers; line 132 and 167 “Fig. 1” refers to Fig. 2; please check throughout the manuscript

Our response: We carefully checked the figure numbering throughout the manuscript and revised it whenever necessary. 

- Line 191 ff:

o inconsistency – sometimes “condition”, sometimes “group” (l.199)

Our response: We now consistently use the term condition throughout the revised manuscript. 

o There is some confusion between text and figure: “condition 2” (line 197) refers to conditions 3 in the figure; and “group 3” (line 199) refers to condition 4 in the figure. Line 325 “Fig. 3” refers to Fig 1. Please check and correct. 

Our response: We are sorry for the inconsistencies in the previous version of the manuscript. We carefully checked the numbering and revised it whenever necessary. 

- 215: please explicitly add (in brackets?) if this condition refers to condition 1 in the figure

Our response: The manuscript has been revised accordingly: “The (digital) passive control condition will be to watch a non-stressful documentary video (condition 1 in Fig 1).” (lines 243-245). 

- Please clarify: non-digital secondary task: how is the VFT instruction given - a written instruction? A verbal instruction provided by the experimenter?

Our response: The manuscript has been revised to “The instruction, which will be verbally given by the experimenter, is to name as many words as possible…” (line 230). 

- Please clarify: digital interruptions: does the primary task stop or does it continue (where – in the background?) when the interruption appears on the same screen?

Our response: The manuscript has been revised accordingly and “The primary task will stop and will be overlayed by the secondary one during the presentation of the secondary task and will automatically continue after a response is given by the participant.” has been added (lines 221-223). 

- Line 276/277: what are “items from 64”? Please clarify.

We clarify this in the revised manuscript: “For assessment of habitual television and internet usage, items from a questionnaire for the assessment of habitual media consumption by Koch (2010; 87) will be used. Each subscale consists of 8 items that should be answered on 7-point Likert-scales. A further scale, which assesses habitual mobile phone usage, has been developed analogously based on the items by 87 for habitual television and internet usage.” (lines 311-315). 

- Why is depression assessed twice – CES-D and STADI-T?

Our response: We will use the CES-D as our main depression-assessment tool We will use the STADI-T primarily for anxiety assessment. In order not to alter the psychometric properties of the STADI-T, we decided to leave in the depression items. We clarify this in the revised manuscript: “Note that we will use the CES-D as main outcome measure for depression. However, we will leave in the depression items to not alter the psychometric properties of the STADI-T.” (lines 350-352). 

- 353: typo: plural – triplicateS

Our response: The typo has been revised (line 443). 

- Randomization/group composition: The authors are experienced when it comes to effects of menstrual cycle phase and hormonal contraceptives on physiological stress reactivity. Please explain how you plan to rule out differential effects of female participants´ follicular and luteal phases of the menstrual cycle on (cortisol) stress reactivity and how to balance group compositions in terms of sex, use of hormonal contraceptives, and menstrual cycle phases.

Our response: We will assess the menstrual cycle phase of the female participants and will consider it in the statistical analysis. Groups will be stratified across their sex distribution: “Participants’ sex will be considered in the randomization process to ensure an equal sex distribution” (lines 466-467). A further stratification with regard to the use of hormonal contraceptives as well as menstrual cycle phase would not be feasible from an organizational point of view. 

Reviewer #2: 

Thank you for giving me the opportunity to review the study protocol regarding the study „Physiological stress in response to multitasking and work interruptions“. The study planned by Linda Becker and coworkers is of interest to the readership of PLOS in my opinion as it deals with a topic that affects many persons during digitalization, namely being confronted with many tasks at the computer at the same time. In the planned study the effects of single tasking, dual tasking and multiple tasking on physiological parameters will be assessed. I think the study proticol needs clarification in some points though. Here are my concerns:

Our response: We would like to thank the reviewer for carefully reading and commenting on our manuscript and for all the valuable suggestions. 

1. Why do you assess IL-6 as this is not a typical stress marker and also you very clear state in the theoretical background that you expect stress effects on the immune systems with delay. I would suggest to measure IL-6, but rather not directly after the tasks.

Our response: We expect fast effects for Il-6 and provide references in the Introduction of the revised manuscript: “With a further delay (e.g., about 1.5- 2 hours for interleukin-6 (IL-6); 23,24), complex effects of the immune system are activated …” (lines 75-76). 

2. Page 5, line 80: Please explain what maladaptibe stress response patterns are in your opinion.

Our response: We refer to “…to stress-responses patterns as being maladaptive – in contrast to adaptive (e.g., 32,33) – when they do not allow the organism to efficiently cope with or to adjust the individual’s physiological responses or behavior to the situation.” And added this to the revised manuscript (lines 81-84). 

3. Page 5, line 84: Please also give some examples on the relationship between stress and dermatological conditions as the relationship is well understood in this field.

Our response: We added the following paragraph to the Introduction “Moreover, these patho-physiological stress-related processes are associated with a large number of other diseases such as chronic dermatological conditions (e.g., skin aging (37), urticaria (38,39),or skin tumors; (40)), asthma bronchiale (41,42), or obesity (43,44), and many more.“ (lines 91-94). 

4. Page 6, line 123: In case the physiological stress response ist associated with these person characteristics, they need to be considered as covariates! Please add!

Our response: We plan to include the factors age, sex, BMI, use of oral contraceptives, and time of day as covariates in the main analyses, because these factors are known to be associated with stress system activity for other stressors (lines 141-144 and 485-487). The further person characteristics (e.g., personality) will be assessed for additional exploratory analyses and will, therefore, not be considered as further covariates. 

5. Page 7, line 134: why do you use such an age restriction?

Our response: We used a pragmatic age restriction of 40 years, because this has been shown to be suitable to recruit healthy participants who do meet all inclusion criteria in previous studies from our group. We clarify this in the revised manuscript and added this as a limitation to the Discussion section: “We will use a pragmatic age restriction of 40 years, because this has been shown to be suitable in previous studies from our group for recruiting healthy participants who do meet all inclusion criteria. Other age groups (older than 40 years as well as children and adolescents) would be interesting target groups for future follow-up studies.” (lines 565-568). 

6. Page 7, line 135: How do you operationalize psychological disorders (ever during life or at the current moment?)

Our response: We operationalize psychological disorders as disorders that have been diagnosed within the last 2 years. We added this to the revised manuscript (line 153). 

7. Page 7, line 136: Please consider to stratify groups for usage of oral contraceptives. I would prefer to do this instead of just statistically controlling for this factor.

Our response: A further stratification – beside sex – with regard to the use of hormonal contraceptives as well as menstrual cycle phase would not be feasible from an organizational point of view, but they will be considered as covariates. We clarify this in the revised manuscript: “Participants’ sex will be considered in the randomization process to ensure an equal sex distribution.” (lines 466-467) and “The potential confounders age, sex, BMI, use of oral contraceptives and menstrual cycle phase for female participants, as well as time of day will be included in all statistical analyses as covariates.”. (lines 485-487)

8. Page 7, line 137: Why is being an employee of the University Erlangen-Nürnberg an exclusion criterion? I suggest to omit or explain further why this is necessary.

Our response: This is a requirement from the workers’ council of our university. This has been added to the revised manuscript (line 156). 

9. Has the study protocol been registered at DRKS? If not please do so.

Our response: The study protocol has not been registered at DRKS yet. Since submission of this study protocol (August 2021), we started data collection (end of September 2021). Therefore, we are not allowed to register the study at DRKS. In our opinion, the publication of this research protocol is a very good alternative for a pre-registration of our study. 

10. Page 8, line 161: What is the content of the non-stressful video. How do you guarantee that the video does not induce relaxation? Would that be a problem? Please think about this and add to the discussion section.

Our response: The content of the video is landscapes and animals: “The (digital) passive control condition will be to watch a non-stressful documentary video (condition 1 in Fig 1). The content of the video is landscapes and animals. These videos have been successfully used in previous studies and evaluated as being non-arousing (65,66).” (lines 243-245). 

Moreover, we added the following paragraph to the Discussion section: “A further potential limitation is the chosen passive control condition as we cannot rule out that watching the videos unintendedly leads to either stress induction or relaxation. However, the videos’ contents have been rated as being low arousing in previous studies (65,66). Moreover, this task is better suited than other potential control tasks in which participants are instructed to do nothing at all, which are – in our opinion – more likely to induce relaxation and which are also much more difficult to control.” (lines 558-564). 

11. Page 9, line 171: Please already explain at this point: Who will be involved in randomization? Are subjects blinded? If yes, what do they think the intention of the study is?

Our response: A team member who is not involved in data collection is responsible for randomization. Subjects are blinded and are informed that the intention of the study is the assessment of physiological responses to interaction with digital devices. This has been added to the revised manuscript: “A team member who is not involved in data collection is responsible for randomization. Subjects are blinded and are informed that the intention of the study is the assessment of physiological responses to interaction with digital devices.” (lines 463-465). 

12. Page 9, line 178: please explain this abbreviation

Our response: We revised the sentence to “We will use an AX-CPT variant (e.g., 61), i.e. that the target is the letter ‘X’ occurring after an ‘A’.’” (lines 200-201)

13. Page 10, line 194: Is intelligence (IQ) used as covariate? Intelligent persons might not get frustrated so fast which can have an effect on the stress parameters. Please consider!

Our response: We only assess IQ in some of the conditions and always in combination with other tasks. Therefore, the results do not necessarily reflect the participant’s ‘real’ IQ. Although an additional comprehensive assessment would be interesting, this would go beyond the scope of our study. We added this to the discussion section: “Furthermore, although we assess intelligence during some of the sub tasks, a comprehensive assessment (including emotional intelligence; 107) would be even more meaningful.” (lines 556-558)

14. Page 11, line 225: IL-6: See above, as stated in the Theoretical background you do not really seem to expect short-term effects on IL-6 or do you? Please, be specific. Do you expect effects after 24 hours or also immediate effects?

Our response: In our experience, IL-6 responses can already be found after 90 minutes after the acute stressors (e.g., McInnis et al., 2014). We, therefore collect the second blood sample at this time point. CRP responses are slower and can be expected about 24 hours later. We, therefore, collect an additional sample at the following day. We now specify our expected time course at lines 413-414 in the revised “Slower responses are expected for the immune system with a maximum of 90 minutes after the task for IL-6 and s-IgA and 24 hours after the experimental session for CRP.”. Furthermore, we revised the Introduction to “With a further delay (e.g., about 1.5- 2 hours for interleukin-6 (IL-6); 23,24), complex effects of the immune system are activated…” (lines 75-76)

15. Page 12, line 257: I suggest to come up with a more detailed figure showing the timeline and in which you illustrate when each variable will be assessed using different symbols for the different variables. Figure 1 is not quite understandable at one glance.

Our response: The figure (no. 3 in the revised version of the manuscript) has been revised accordingly. 

16. Page 13, line 264 and the following: what will you do with all these variables? Do you plan to use them as control variables? Are these dependent variables? If yes, which correction procedure (adjustment) will you use?

Our response: We will use these variables for exploratory analyses to investigate research question 5. They will not be included as control variables. For all these analyses, an adjusted alpha-level of 0.001 will be used (line 496). 

17. Page 17: What is your main heart parameter? It seems that you will exploratively look at everything regarding the heart rate variability. Please, explicitely state at this point whether you will correct for multiple testing or not.

Our response: We will use the RMSSD as the main HRV parameter (lines 448-449). We will correct for multiple testing when we exploratively investigate the other parameters and will correct for multiple testing and an adjusted alpha-level of .001 will be used (lines 488-459)

18. It seems a bit odd to have a results section in study protocol, thus of a study which is still ongoing or has not even started yet. Please, omit this heading and include the necessary information of this section somewhere else (Methods Section or Theoretical background).

Our response: We revised the manuscript accordingly and included the paragraphs from the Results section into the Methods section and also removed it from the Abstract. 

19. Has data collection started already?

Our response: Data collection has started in September 2021 after submitting the first version of this study protocol. This has been added to the revised manuscript: “Recruitment has started in August 2021 and data collection has started at the end of September 2021. The first version of this study protocol has been submitted before start of data collection. All recommendations from the reviewers’ first reviews were considered.” (lines 173-175). 

20. Is it planned to conduct inbetween analyses?

Our response: In-between analyses are not planned. This has been added to the revised manuscript: “Interim analyses are not planned..” (lines 506-507). 

21. Please add a data management plan. Where will the data be stored, who will have access, will the data be made available to other researchers or in data repositories? If not, please give an explanation. 

Our response: We added a data management plan to the Methods section in the revised manuscript: “To protect the participants’ privacy and to maintain confidentiality, all personal data is stored in password-protected files and secured against unauthorized access by third parties. The raw data and materials are only accessible to project-team members. Each participant is assigned a randomly generated code that does not allow any conclusions to be drawn about the person. Only this code is used for naming files and samples. Only completely anonymized data will be made available to other researchers after completion of the study or in data repositories. Only mean values and group statistics will be reported in publications.” (lines 498-504). 

22. What is the clinical relevance of the study? Please add to the discussion session.

Our response: We included a paragraph about the implications of our study (including the clinical relevance) at the end of our discussion section in the revised manuscript: “Nevertheless – despite these limitations –, we deem that our study contributes to a deeper understanding on influences of stressors associated with the use of digital technology on humans' health-outcomes. Specifically, our results are expected to expand our current knowledge base on the impact of multitasking and information load on humans' psychobiological stress responses with particular focus on physiological response patterns. Stress exerts a strong influence on health via well-described processes (e.g., conceptualized in the Allostatic Load Model; 3,32,110). In the long-term, stress is negatively associated with quality of life, health, and longevity of individuals and, thus, productivity of society (111,112). Therefore, our study is of high relevancy. Given the ubiquitous applications of digital technologies in modern workplaces and living environments, our findings will help to further understand the mechanisms between digital stressors in various occupational settings and adverse health outcomes (113). Eventually, examination of job-related risks will inform policy and practice interventions in occupational health. Overall, the findings from our study will have important implications for better understanding the long-term health effects of the potential stressors multitasking and work interruptions in several settings.” (lines 573-588).

---

## [Decision Letter · Decision Letter 1]

27 Jan 2022

Physiological stress in response to multitasking and work interruptions: Study protocol

PONE-D-21-28194R1

Dear Dr. Becker,

We’re pleased to inform you that your manuscript has been judged scientifically suitable for publication and will be formally accepted for publication once it meets all outstanding technical requirements.

Kind regards,

Eva M J Peters, M.D.

Academic Editor

PLOS ONE

Additional Editor Comments (optional):

We are happy to accept your revised manuscript. Please attend to the last concerns raised by the reviewers.

Reviewers' comments:

Reviewer's Responses to Questions

**Comments to the Author**

1. Does the manuscript provide a valid rationale for the proposed study, with clearly identified and justified research questions?

Reviewer #1: Yes

Reviewer #2: Yes

2. Is the protocol technically sound and planned in a manner that will lead to a meaningful outcome and allow testing the stated hypotheses?

Reviewer #1: Yes

Reviewer #2: Yes

3. Is the methodology feasible and described in sufficient detail to allow the work to be replicable?

Reviewer #1: Yes

Reviewer #2: Yes

4. Have the authors described where all data underlying the findings will be made available when the study is complete?

Reviewer #1: Yes

Reviewer #2: Yes

5. Is the manuscript presented in an intelligible fashion and written in standard English?

Reviewer #1: Yes

Reviewer #2: Yes

6. Review Comments to the Author

You may also provide optional suggestions and comments to authors that they might find helpful in planning their study.

Reviewer #1: Congratulations to the authors for this very nice revision. My comments have been fully addressed. I have two last very minor comments:

1. Line 140: One bracket is missing after “…(so-called polychronicity; 57))

2. Line 315: just listing a number (“87”) without brackets and thus a clear indication that it refers to a literature reference is irritating – please clarify by e.g. adding the author “… has been developed analogously based on the items by Koch (87) for...”.

I recommend to accept the manuscript for publication.

Reviewer #2: The study protocol has been improved according to my suggestions and I am fine with the study protocol in its current form.

7. PLOS authors have the option to publish the peer review history of their article (what does this mean?). If published, this will include your full peer review and any attached files.

Reviewer #1: No

Reviewer #2: No

---

## [Editor Report · Acceptance letter]

31 Jan 2022

PONE-D-21-28194R1 

Physiological stress in response to multitasking and work interruptions: Study protocol 

Dear Dr. Becker:

I'm pleased to inform you that your manuscript has been deemed suitable for publication in PLOS ONE. Congratulations! Your manuscript is now with our production department. 

Kind regards, 

on behalf of

Prof. Dr. Eva M J Peters 

Academic Editor

PLOS ONE